# Aggregated Individual Reporting for Post-Deployment Evaluation: Mechanism Design & Modeling Considerations

## Abstract

Evaluating the real-world behavior of AI systems once they are deployed is a key component of understanding their societal impact. One approach proposed in recent work is *aggregated individual reporting* (AIR), where end-users of deployed systems are able to submit feedback ("reports") to a central mechanism, which aggregates these reports to compute an evaluation of the deployed system [Dai et al., 2025b]. The goal of the mechanism is to understand the true state of the world, based on submitted reports. A key open question is how an optimal AIR mechanism might be designed when reporting behavior is taken explicitly into consideration. This gives rise to two simultaneous challenges: First, that of designing rewards for reporting in order to incentivize high-quality feedback, and second, that of making reliable (statistical) inferences on an inherently "non-i.i.d." sample of information. In this extended abstract, we describe some work-in-progress that seeks to initiate rigorous study of these problems. We provide a "maximalist" model of the interaction between end-users and an AIR mechanism, as well as some "minimal" example instantiations of the model. We discuss various research questions that naturally arise from this model.

## 1  Introduction

End-user feedback is a promising source of information about the real-world performance of deployed AI systems. On the one hand, it is impossible for model developers to anticipate the full scope and depth of how their systems will be used; on the other, the individuals who directly experience these AI systems have unique perspectives on their impacts. To this end, mechanisms for *aggregated individual reporting* (AIRs), initially studied in Dai et al. [2025a] and discussed further in Dai et al. [2025b],seek to utilize reports from end-users of deployed systems as a source of data for constructing evaluations of these systems. The hope is that end-user feedback can identify "unknown unknowns" in system behavior, while algorithmic aggregation can concretize anecdotal experience as statistical evidence. For example, Dai et al. [2025a] proposes a method for post-deployment fairness auditing from individual reports by formalizing the problem as a sequential hypothesis test, thereby converting a collection of "individual experiences" into something statistically legible as "collective evidence."

More generally, Dai et al. [2025b] defines an AIR as a mechanism that satisfies the following: (1) *individual reporting* (reports are submitted by end-users about specific experiences with the evaluated system); (2) *aggregation for evaluation* (reports are aggregated and interpreted over time, with the goal of evaluation); and (3) *evaluation-conditional action* (there are evaluation outcomes where, if and when the reports are consistent with those outcomes, downstream action can be taken). AIRs have the potential for a wide range of applications, and to identify real-world "unknown unknown" safety problems—we reproduce Figure 1 from Dai et al. [2025b] in the Appendix, as an illustration—yet further study remains necessary.

A crucial open question that remains unresolved is *how* such a mechanism ought to be designed when taking the behavior of individual reporters into account. For example, what would motivate an end-user to submit a report, and how does that affect the content of reports received by the mechanism? When end-users are able to coordinate their reporting behavior, such as via social media, does this help or hinder the mechanism's ability to make accurate inferences about the true impact of the deployed system? How are reports affected by expectations about the nature of the downstream action, and about the nature of other end-users' behavior?

In this work, we take initial steps towards understanding these questions. The goal is to design a mechanism for soliciting, interpreting, and rewarding individual reports so that a high-quality evaluation of a deployed system can be computed from reports in aggregate; however, doing so requires developing a more formal understanding of including the various ways that individuals might (be incentivized to) interact with a mechanism, and with one another. To that end, the remainder of this extended abstract highlights potential modeling options and design decisions for the study and development of AIR mechanisms.[1]

## 2  Warm-up: Modeling assumptions of prior work

To contextualize the challenges to be addressed, we begin with a summary of the modeling assumptions required in Dai et al. [2025a]. In this work, the goal was to identify issues of unfairness, as described by disproportionate rates of harm experienced by a particular subgroup in the population.

Specifically, all individuals belong to at least one group $G \in \mathcal{G}$, and group membership for individual $i$ is determined entirely by their feature vector $X_i$; for each individual, whether or not they experience (ground-truth) "harm" is an unobserved random variable $Y_i \in \{0, 1\}$. If an individual decides to report, their feature vector $X_i$ becomes visible to the mechanism; the event of a report being submitted is denoted $R_i \in \{0, 1\}$. The likelihood of observing a *report* at any given time for each group $G$ is controlled by $\mu_G := \Pr[X \in G \mid R]$.

The goal, as described above, can thus be formalized as determining the degree to which the frequency of harm for any group $G \in \mathcal{G}$, i.e., $\Pr[Y \mid G]$, exceeds the population average $\Pr[Y]$; the ratio $\frac{\Pr[Y|G]}{\Pr[Y]}$ is referred to as the *relative risk* for $G$. This goal is achieved by determining whether the frequency of *reports* from any group $G$, exceeds a baseline relative to its base rate in the population, notated as $\mu_G^0$—that is, $\mu_G \geq \beta \mu_G^0$ for some $\beta > 1$.

Notably, Dai et al. [2025a] does not model the decision of individuals to report—i.e., how $R$ is realized, how $\Pr[R \mid G]$ is determined, or how $R_i$ (the event of an individual reporting) is related to $Y_i$ (the event of the same individual experiencing harm) directly. Instead, Proposition 3.2 of Dai et al. [2025a] claims that it is sufficient to assume that $\frac{\Pr[R|G]}{\Pr[Y|G]}$ is not too different from $\frac{\Pr[R]}{\Pr[Y]}$ (e.g. by a factor of $b$); under this assumption, the true relative risk for $G$ compared to the population average is at least $\frac{\Pr[Y|G]}{\Pr[Y]} \geq \beta/b$.

While simple, this model already identifies several key components of what a more general model of AIRs must include. First, the ground-truth state of the world, which involves the population interacting with the deployed system, as well as their experiences of the system; second, the mechanism itself, which involves not just the aggregation algorithm and the information that reports can contain, but also what rewards or incentives can be offered to individuals who report; and finally, the reporting behavior of the overall population, which involves decisions about whether to report, and what information to submit within the report. We now turn to formalizing these components more explicitly.

## 3  A maximalist model for AIRs

We begin by introducing a model that enumerates a wide range of considerations that may become relevant for the study of AIRs. Realistically, concrete insights (e.g., the potential research questions discussed in Section 4) will require focusing on more specific components of the model; however, we provide this "maximalist" model for completeness, and to illustrate the range of relevant questions.

---

[1]Our submission to this workshop is a snapshot of work-in-progress, with the goal of highlighting potential future directions that may be of interest to this community. We defer a literature review—and, of course, quantitative results from working with variations of the model—to a later version of this manuscript.

**I. The deployed system, the relevant population, and its impacts.** Regardless of the existence of a reporting mechanism, the ground-truth state of the world depends only on the relevant population interacting with the deployed system and their experiences with the system. A key component of this "ground-truth" state is individuals' subjective *perception* of particular experiences.

*Population.* Individuals interacting with the deployed system can be parametrized with covariates (e.g., demographic, task, prompt, etc.) from feature space $\mathcal{X}$. The population of these individuals can be modeled with distribution $\mathcal{D}_X$, where features for any individual $i$ are drawn $X_i \sim \mathcal{D}_X$.

*Experiences.* The space of possible experiences (for example, prediction or allocation outcomes, transcripts of model responses, or summaries of behavior) from interacting with the deployed system is denoted $\mathcal{Y}$. $\mathcal{D}_Y$ is the marginal distribution of experiences induced over the population, and $\mathcal{D}_{Y|X}$ is the distribution conditioned on covariates. For an individual with covariates $x_i$, their experiences are drawn $Y_i \sim \mathcal{D}_{Y|x_i}$.

*Perceptions.* For each individual $i$, we use $\iota_i \in [-1, 1]$ as a scalar value that captures some subjective measure of the (dis)utility perceived by individual $x_i$ for experience $y_i$. Distributional or smoothness assumptions can also be made here.

**II. The AIR mechanism.** The evaluation mechanism $\mathcal{M}$ takes (a sequence of) reports and returns some summary or identification of the most significant issue raised in those reports. After this issue is identified, various types of downstream action may be taken—e.g., as illustrated in Figure 1, investigation of the identified issue, improvement of the deployed system, future policy changes about usage guidelines, etc. One key observation is that the identification of the issue goes through a set of "important reports" that are, in some way, representative of the significant issue. It is therefore this same set of "important reports" that receive the benefits of having not only reported but also having their reports correspond to the important issue.

*Report content.* A report $r_j$ is some function of the tuple $(x_j, y_j, \iota_j)$. For example, one possibility is that every report directly and truthfully reports all covariates, experiences, and perceptions, i.e. $r(x_j, y_j, \iota_j) = (x_j, y_j, \iota_j), \forall j$. A mechanism may not make it possible to report all items in the tuple; e.g., a mechanism may only allow submissions to include $\mathcal{Y}$.

*Administrator knowledge.* The framework introduced in Dai et al. [2025b] distinguishes between several types of "mechanism administrators," depending on their relationship to the deployed system. This distinction is relevant here primarily regarding what kind of additional information $\mathcal{M}$ may have access to.

Generally, $\mathcal{M}$ only has access to the reports that are actually submitted. However, if $\mathcal{M}$ is "first-party," i.e., owned by the same organization as the deployed system, then it would have access to the "experiences" $y_i$ from all users, not just reporters (i.e., it has access to realizations of samples from the marginal $\mathcal{D}_Y$), though it may not have full knowledge of $\mathcal{X}$ or $\mathcal{D}_X$. It may also have the ability to query for information about $x_i$ and $\iota_i$ from the full population.

*Aggregation and evaluation.* Given $T$ total reports $\{r_j\}_{j \in [T]}$, the mechanism computes an evaluation for the deployed system. Each report has a real-valued "importance" to the evaluation, $\mathcal{I} \in [0, 1]^T$. For example, if the evaluation is computed via a clustering algorithm and finding a most-significant cluster, then $\mathcal{I}_t$ can capture $r_t$'s (normalized) distance to the center of the cluster deemed most significant; on the other hand, if the evaluation simply identifies a subset of reports, as in Dai et al. [2025a], then $\mathcal{I} \in \{0, 1\}^T$, and $\mathcal{I}_t$ captures whether $r_t$ belonged to the relevant subgroup identified.

*Rewards/incentives/decisions.* We can abstract the types of benefit that a mechanism might provide into three categories. First, there may be some inherent benefit to reporting, regardless of how important the report ends up being to the final evaluation, e.g. a per-report response or reward provided by the mechanism; we denote this $p_r$. Second, there is some benefit that is afforded to reports depending on their importance to the evaluation, e.g. a direct reward to reporters that scales with relevance; we denote this $p_{\mathcal{I}}$. Finally, there may be a benefit afforded to individuals similar to those who reported, but who did not actually report, e.g. the deployed model improving for a specific subset of tasks/covariates; we denote this $p_{\widetilde{\mathcal{I}}}$.

**III. Reporting behavior.** With the "ground-truth" and mechanism in place, we are now equipped to reason about *when* an individual $i$ would report, and *what* they would report.

**Costs of reporting.** We can think of the costs of reporting as including some fixed term for all reporters $c_0$, and some covariate-dependent term that captures variation in difficulty across the population $c(X)$. These terms could be (e.g.) additive, so that the cost for an individual $j$ is $c_j = c_0 + c(x_j)$.

**(Expected, counterfactual) benefits of reporting.** The overall benefits of reporting depend not only on the potential payments made by the mechanism $(p_r, p_{\mathcal{I}}, p_{\widetilde{\mathcal{I}}})$, but also the individual's subjective perceptions of their experiences and their beliefs about other reporters.

First, we can think of $u_r$, the subjective analogue to $p_r$, as a quantity that depends only on $\iota$ (the individual's subjective rating of their experience's importance or intensity); if they felt strongly about their experience, they may derive satisfaction or relief from submitting a report about it, regardless of expectations about whether their report would be addressed.

Second, how would an individual $(x_j, y_j, \iota_j)$ would reason about their *expected* benefits, based on $p_{\mathcal{I}}$ and $p_{\widetilde{\mathcal{I}}}$? The key quantity is the individual's *belief* about the likelihood that their report would be deemed "important," if they submitted a report $\Pr[\mathcal{I}_j = 1 \mid j \text{ reports}]$, and about the likelihood that the experience that they are concerned about would be deemed "important," if they did not report.[2]

All together, the expected benefit of reporting is $\mathbb{E}[b_j] = p_{\mathcal{I}} \cdot \Pr[\mathcal{I}_j = 1 \mid j \text{ reports}] + u_r + p_r(\iota_j)$, while the expected benefit of not reporting is $\mathbb{E}[\widetilde{b}_j] = p_{\widetilde{\mathcal{I}}} \cdot \Pr[\widetilde{\mathcal{I}}_j = 1 \mid j \text{ doesn't report}]$.[3] We expect that the question of how potential reporters form these beliefs is a core challenge for this line of work.

**Reporting likelihood.** A basic assumption is that individuals would submit a report only if their expected benefits of submitting a report exceed their perceived costs of submitting a report; we will denote this quantity $\Delta_j := \mathbb{E}[b_j] - \mathbb{E}[\widetilde{b}_j] - c_j$. The *likelihood* that an individual $j$ submits a report scales with the degree to which their benefits outweigh their costs; e.g., one such function might be $1 - \exp(-\Delta_j)$, possibly scaled by a global constant to account for overall low rates of reporting.

**Reporting content.** Finally, given that individual $j$ has decided to report, what would they actually submit? For example, suppose that every report $j$ observed by $\mathcal{M}$ is a noisy signal of $j$'s true experience, so that $r_j = y_j + \eta_j$. One further way in which the degree of benefit/cost gap can have an impact is the willingness to expend *effort* to submit a higher-quality report, which can be modeled as lower noise. For example, $\eta_j \sim \mathcal{N}(0, \mathbf{I} \cdot (1 - \Delta_j))$.

# 4 Minimalist instantiations & example research questions

We conclude with a brief discussion of example research directions based on this model.

**The impact of reporting data on reliable inference.** In the absence of tailored rewards/incentives for reporting, what would one expect the distribution of reports to look like, and is it possible to directly correct for this when making inferences? For example, suppose the population is homogeneous, and experiences are drawn $y \sim \text{Bin}(p)$. $\mathcal{M}$ does not know $p$, but wants to estimate $\widehat{p}$. How do different joint distributions of $(y, \iota)$ affect the estimation $\widehat{p}$? If $\mathcal{M}$ instead is concerned about whether $p$ is greater than some threshold $\tau$, what changes in reporting behavior (and therefore available data) does this changed objective induce?

**The potentials and limitations of coordinated (collective) action.** Since a core component of an AIR is the aggregation, the presence of coordination among reporters can dramatically affect the set of feedback that is observed by the mechanism, especially as decisions about reporting depend heavily on individual beliefs about the possibility of their report making an impact. What kinds of information should the mechanism publicize, and what should remain private? Can implicit cooperation arise by, e.g., publicizing intermediate results of evaluation, and how can coordination be encouraged explicitly, as a means to identify the highest-priority problems?

**The "marginal benefit" of deploying an AIR.** As an organization that owns the deployed system ("first-party"), the organization is able to sample or query for feedback over the full population of users directly. Under what conditions is doing so preferable to running a reporting mechanism? For instance, since we expect that data from reports is non-representative, what is the tradeoff between data quality and data cost? Is there an optimal way of using a reporting mechanism (e.g. as a seed for further information collection)?

---

[2]For ease of exposition, suppose importances are binary.

[3]Here, the event $\{\widetilde{\mathcal{I}}_j = 1\}$ indicates the event that the experience $j$ cared about was identified via the submitted reports.

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

# A   Omitted figure

| Evaluated system | Mechanism administrator | Affected population | Individual report information | Evaluation condition (when would downstream action be taken due to patterns in reports?) | Downstream actions |
|---|---|---|---|---|---|
| **FDA-approved vaccines** (deployed in real-world system as VAERS) | 3rd-party (United States CDC & FDA) | All patients who received a particular vaccine | Specific vaccine (brand and batch), specific adverse event (e.g., ) and demographic information | Elevated overall frequency of adverse event reports compared to expected baseline frequency *e.g., myocarditis appears frequently for the COVID-19 vaccine.* | Further investigation of specific vaccine-side effect pairs (e.g. additional research or data collection), and notification of relevant parties (e.g. published reports) |
| **Loan allocation algorithm at Bank X** (hypothetical from Dai et al. 2025) | 3rd-party (activist organization) | All loan applicants to Bank X | Demographic information and the claim of potential discrimination | Identification of a subgroup that experiences disproportionate rates of harm *e.g., financially-healthy Black applicants are denied loans at a higher rate.* | Gathering evidence to initiate a legal discrimination case |
| **AI medical scribe product** (speculative example) | 2nd-party (hospital system client) | Healthcare workers who use the tool, and their patients | Free-text notes and information about reporter; scribe text and edit history, for healthcare worker reporters | Clinically-relevant failure modes of the scribe product. *e.g., AI scribe repeatedly makes errors for visits about pregnancy complications.* | Feedback provided to company developing AI scribe; temporary usage guidelines given to clinicians working in (e.g.) maternal health |
| **ChatGPT** (speculative example) | 1st-party (OpenAI) | All users of the ChatGPT product | Chat transcript and free-text notes | Wide-scale safety-critical behavior. *e.g., the newest model exhibits dangerously sycophantic behavior.* | Rollback to prior model version; post-mortem conducted for flaws in internal evaluations. |

Figure 1: *Examples of how AIRs could be set up for a variety of applications; reproduction of Figure 2 from Dai et al. [2025b].*

