# OpenReview forum: "Aggregated Individual Reporting for Post-Deployment Evaluation: Mechanism Design & Modeling Considerations"
_NeurIPS.cc/2025/Workshop/Reliable_ML — NeurIPS 2025 - Reliable ML Workshop_

### Official Review · Reviewer_Ym7P · 2025-09-18
**Blueprint for strategic AIR but feels very preliminary**

**Rating:** 4
**Confidence:** 3

**Review:**

**Summary:**
- This extended abstract describes some ideas for extending the aggregated individual reporting (AIR) model. In the AIR model, agents submit reports to a central operator, who then processes these reports for a downstream task. The authors describe several ideas for modeling the strategic behavior of agents by characterizing their benefits and costs associated with reporting.

**Strengths:**
- The work does a good job in Sec 2.1 of explaining the previous work of Dai et al. [2025], which established the AIR model.
- The AIR framework seems quite relevant, especially in the context of LLMs, where these mechanisms often ask for feedback about model outputs.
- The research direction on collective action feels worthy of pursuing and likely has a lot of interesting questions. I like the idea that it might actually be a good idea to encourage coordination as a tool to find high-priority problems.

**Weaknesses:**
- This extended abstract feels too preliminary and seems to rely heavily on two papers by Dai et al.
- While there is a comment that this is a work-in-progress, I am disappointed that there is no related works section or quantitative results.

**Suggestions:**
- It would be an interesting question to understand not only whether a person would report or not report, but whether they would strategically misreport to boost relevance, influence on model outcome, etc.

---

### Official Review · Reviewer_u3oS · 2025-09-20
**An research agenda for post-deployment auditing mechanism for AI systems, but requires more polishing**

**Rating:** 4
**Confidence:** 3

**Review:**

## Summary

This paper proposes potential research formulations and directions on top of the Aggregated Individual Reporting (AIR) mechanism, which is a post-deployment auditing mechanism for AI systems. In the AIR mechanism, users submit individual reports, which will be aggregated, and when conditions are met, downstream actions should be taken. An open question is how to take reporting behavior into account. And there are challenges such as incentive design to encourage high-quality feedback and the non-i.i.d. distribution of reports.

The authors first present a “maximalist” model that enumerates:
1. The parties involved: population, experiences, perceptions.
2. The mechanisms: report content, administrative knowledge, aggregation and evaluation, and rewards/incentives/decisions.
3. The reporting behavior: cost, benefit, likelihood, content.

Then, the authors sketch three “minimalist” instantiations:
1. Reliable inference without incentives.
2. Coordination among reporters.
3. Marginal benefit of AIR.

There are no empirical experiments as this is a conceptual agenda and prospectus.

## Strengths

1. **Timely and relevant to workshop topic**. How to design an audit system for AI systems is an open question. User reports are inherently noisy, non-i.i.d., and scarse in some aspects. This matches the topic of the workshop.
2. **Comprehensive modelling**. In the maximalist model, the authors made a structured decomposition of elements involved in the audit system.
3. **Clearly structured**. This paper is modular and highly readable.

## Weaknesses

1. **Lack of theoretical and empirical results**. This paper does not provide new theorems or empirical results. Although it is fine for a workshop paper, some minimal results would definitely add to its rigor.
2. **Lack of justification**. Most of the proposed formulations do not accompany justification or comparison with alternatives, making the feasibility unclear to readers.

## Suggestions for Authors

1. **Aim for a minimal demonstration**. Start with any one of the minimal instantiations to provide possible insight to the community.
2. **Add empirical justification whenever possible**. Accompany proposed formulations with empirical demonstrations in existing systems to justify the reason behind them.